# Mixed topological semimetals driven by orbital complexity in two-dimensional ferromagnets

Chengwang Niu[1,2,6], Jan-Philipp Hanke [2,3,6], Patrick M. Buhl [2], Hongbin Zhang[4], Lukasz Plucinski[5], Daniel Wortmann[2], Stefan Blügel [2], Gustav Bihlmayer [2] & Yuriy Mokrousov [2,3]

The concepts of Weyl fermions and topological semimetals emerging in three-dimensional momentum space are extensively explored owing to the vast variety of exotic properties that they give rise to. On the other hand, very little is known about semimetallic states emerging in two-dimensional magnetic materials, which present the foundation for both present and future information technology. Here, we demonstrate that including the magnetization direction into the topological analysis allows for a natural classification of topological semimetallic states that manifest in two-dimensional ferromagnets as a result of the interplay between spin-orbit and exchange interactions. We explore the emergence and stability of such mixed topological semimetals in realistic materials, and point out the perspectives of mixed topological states for current-induced orbital magnetism and current-induced domain wall motion. Our findings pave the way to understanding, engineering and utilizing topological semimetallic states in two-dimensional spin-orbit ferromagnets.

[1] School of Physics, State Key Laboratory of Crystal Materials, Shandong University, 250100 Jinan, China. [2] Peter Grünberg Institut and Institute for Advanced Simulation, Forschungszentrum Jülich and JARA, 52425 Jülich, Germany. [3] Institute of Physics, Johannes Gutenberg University Mainz, 55099 Mainz, Germany. [4] Institute of Materials Science, Technische Universität Darmstadt, 64287 Darmstadt, Germany. [5] Peter Grünberg Institut, Forschungszentrum Jülich and JARA, 52425 Jülich, Germany. [6] These authors contributed equally: Chengwang Niu, Jan-Philipp Hanke. Correspondence and requests for materials should be addressed to C.N. (email: c.niu@sdu.edu.cn)

Two-dimensional (2D) materials are in the focus of intensive research in chemistry, materials science, and physics, owing to their wide range of prominent properties that include superconductivity, magnetotransport, magneto-, and thermoelectricity. The observations of quantum Hall and quantum spin Hall effects are manifestly associated with 2D materials, and they ignited comprehensive research in the area of topological condensed matter, resulting in the discovery of topological insulators (TIs) and topological crystalline insulators (TCIs) both in 2D and in three spatial dimensions (3D)[1–3]. Recently, research in the area of topological materials has extended to the class of topological semimetals[4–6], which notably include Dirac[7–9], Weyl[10–12], and nodal-line semimetals[13–15]. These materials have been theoretically proposed and experimentally confirmed in 3D, revealing remarkable properties such as ultrahigh mobility[16], anomalous magnetoresistance[17,18], and nonlinear optical response[19]. However, in 2D films the material realization of topological semimetals has been elusive so far[4–6]. Although in some situations a gap closing was argued to occur due to symmetries leading to the realization of 2D Dirac and nodal-line states, a gap is usually introduced once the spin-orbit coupling (SOC) comes into play[20–24].

While magnets have been successfully fabricated in 2D[25,26], combining 2D magnetism with non-trivial topological properties holds great opportunities for topological transport phenomena and technological applications in magneto-electric, magneto-optic, and topological spintronics[27–30]. Thus, studying the unique interplay of topological phases with the dynamic magnetization of solids currently matures into a significant burgeoning research field of condensed-matter physics[30–33]. In this context, magnetic interfaces with topological insulators[34] and layered van der Waals crystals[35,36], which can exhibit ferromagnetism at room temperature, constitute compelling and experimentally feasible classes of 2D quantum materials.

Here, we demonstrate the emergence of zero-dimensional and one-dimensional semimetallic topological states, which arise at the boundary between distinct topological phases when the direction of the magnetization in a 2D magnet is varied. We show that by including the direction of the magnetization into the topological analysis, one arrives at a natural classification of such mixed Weyl and nodal-line semimetallic phases, which paves the way to scrutinizing their stability with respect to perturbations. We uncover that the appearance of semimetallic phases is typically enforced by the drastic variation of the orbital band character upon changing the magnetization direction, which arises commonly in 2D ferromagnets, and we proclaim that emergent semimetals can be experimentally detected by measuring the current-induced orbital response, e.g., via XMCD. Besides providing realistic material candidates in which the discussed semimetals could be observed, we suggest possible applications of these states in shaping the magnetic properties of the edges and current-induced domain-wall motion.

## Results

**Nodal points and lines in mixed topological semimetals.** Topological phase transitions constitute a pervasive concept that necessitates the occurrence of metallic points in the electronic structure. Acting as prominent microscopic sources of geometrical curvature of momentum space, such band crossings are currently discussed in three-dimensional Weyl semimetals, where they mediate a plethora of fascinating properties[10–12]. Analogously, it was suggested[30] that large magneto-electric effects in two-dimensional ferromagnets coined mixed Weyl semimetals (MWSMs) originate from emergent nodal points in the mixed phase space of the crystal momentum $\mathbf{k} = (k_x, k_y)$ and the magnetization direction $\hat{\mathbf{m}}$ (see Fig. 1b). Discovering material candidates and advancing our understanding of topological states in this novel class of semimetallic systems is invaluable for the interpretation of physical phenomena that root in the global properties of the underlying complex phase space.

The emergence of nodal points in MWSMs correlates with drastic changes in the mixed topology and it is accompanied by discrete jumps of the momentum Chern number $\mathcal{C} = 1/(2\pi)\int\Omega_{xy}^{\mathbf{kk}}dk_xdk_y$ with respect to the magnetization direction, as well as of the mixed Chern number $\mathcal{Z} = 1/(2\pi)\int\Omega_{yx}^{\widehat{\mathbf{mk}}}dk_xd\theta$ with respect to the crystal momentum[30]. Here, the momentum

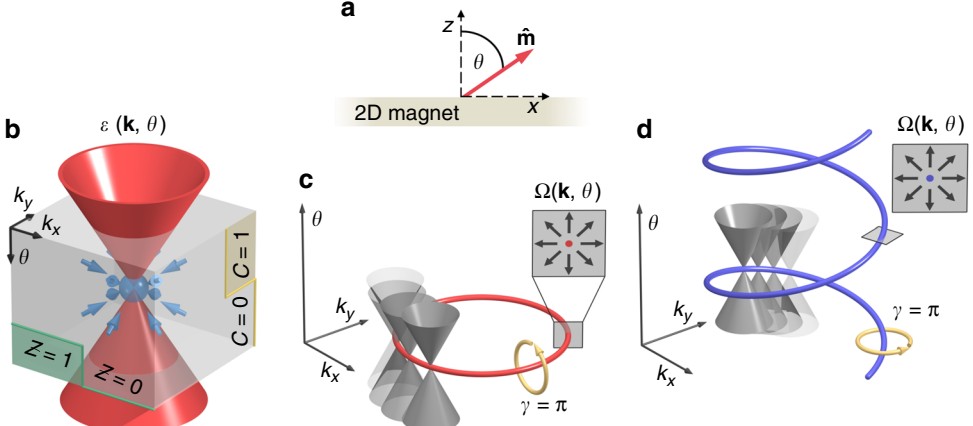

**Fig. 1** Characteristics of mixed topological semimetals. **a** The magnetization direction $\hat{\mathbf{m}} = (\sin\theta, 0, \cos\theta)$ of a two-dimensional magnet encloses the angle $\theta$ with the z-axis perpendicular to the film plane. **b** Acting as sources or sinks of the Berry curvature, emergent band crossings in the mixed phase space of crystal momentum $\mathbf{k} = (k_x, k_y)$ and $\theta$ can be identified with jumps of the momentum Chern number $\mathcal{C}$ and the mixed Chern number $\mathcal{Z}$ upon passing through the nodal points. Alternatively, the topological nature of such a mixed Weyl point can be confirmed by calculating its charge as the flux of Berry curvature through the closed surface indicated by the grey box. **c** If the magnetic system is symmetric with respect to reflections at $z = 0$, nodal lines with the Berry phase $\gamma = \pi$ may manifest in the corresponding $(k_x, k_y)$-plane of the mixed phase space. The inset illustrates the distribution of the generalized Berry curvature field $\Omega$ around the nodal line. **d** Mixed topological semimetals can host additionally a very distinct type of nodal lines that are one-dimensional manifolds evolving in $\theta$ as well as in $\mathbf{k}$. Originating from the complex topology in the mixed phase space as revealed by a non-trivial Berry phase $\gamma$, these nodal lines give rise to a characteristic distribution of the Berry curvature as exemplified in the inset

Berry curvature of all occupied states $|u_{\mathbf{k}n}^\theta\rangle$ is denoted by $\Omega_{xy}^{\mathbf{kk}} = 2\,\mathrm{Im}\sum_n^{\mathrm{occ}} \langle \partial_{\mathbf{k}_x} u_{\mathbf{k}n}^\theta | \partial_{k_y} u_{\mathbf{k}n}^\theta \rangle$, the mixed Berry curvature is $\Omega_{yi}^{\widehat{\mathbf{mk}}} = 2\,\mathrm{Im}\sum_n^{\mathrm{occ}} \langle \partial_\theta u_{\mathbf{k}n}^\theta | \partial_{k_i} u_{\mathbf{k}n}^\theta \rangle$, and $\theta$ is the angle that the magnetization $\widehat{\mathbf{m}} = (\sin\theta, 0, \cos\theta)$ makes with the $z$-axis as depicted in Fig. 1a. To fully characterize the properties of nodal points in the composite phase space spanned by $k_x$, $k_y$, and $\theta$, we introduce the integer topological charge

$$Q = \frac{1}{2\pi}\int_S \mathbf{\Omega} \cdot d\mathbf{S}, \tag{1}$$

which describes the non-zero flux of the generalized Berry curvature field $\mathbf{\Omega} = (-\Omega_{yy}^{\widehat{\mathbf{mk}}}, \Omega_{yx}^{\widehat{\mathbf{mk}}}, \Omega_{xy}^{\mathbf{kk}})$ through a closed surface $S$ that encompasses the nodal point (see Fig. 1b). We classify in the following two different types of such mixed Weyl points in the composite phase space: First, the symmorphic combination of time reversal and mirror symmetries can enforce topological phase transitions accompanied by a closing of the band gap as the magnetization direction is varied. As we discuss below, such type-(i) nodal points are robust against perturbations that preserve the protective symmetry, as long as the magnetization direction is fixed. Second, generic band crossings may arise due to the complex interplay of exchange interaction and SOC in systems of low symmetry. In this case, when the underlying electronic structure is modified, such type-(ii) nodal points disappear as long as the direction of the magnetization is fixed, but they reappear if the magnetization direction is adjusted.

In addition, as we demonstrate below, nodal points in mixed topological semimetals can form closed lines in the higher-dimensional phase space of momentum and magnetization direction (see Fig. 1c, d). It is tempting to interpret these one-dimensional manifolds of topological states as mixed nodal lines in analogy to their conventional sisters in three-dimensional topological semimetals[13–15]. While crystalline mirror symmetry underlies the emergence of the nodal line in momentum space shown in Fig. 1c, mixed topological semimetals host additionally a distinct type of nodal lines as depicted in Fig. 1d. Owing to the subtle balance of spin-orbit and exchange interactions, these

topological states can be thought of as series of nodal points that evolve also with the magnetization direction $\theta$. As a direct consequence, this type of mixed nodal line is not protected by crystalline symmetries but stems purely from a non-trivial Berry phase $\gamma = \oint_c \mathbf{A} \cdot d\ell$, where $\mathbf{A} = i\sum_n^{\mathrm{occ}} \langle u_{\mathbf{k}n}^\theta | \nabla u_{\mathbf{k}n}^\theta \rangle$ is the generalized Berry connection in the complex phase space, $\nabla$ stands for $(\partial_{k_x}, \partial_{k_y}, \partial_\theta)$, and the closed path $c$ encircles the nodal line as shown in Fig. 1d.

**Model of a mixed Weyl semimetal.** To establish the existence of the predicted mixed topological semimetals, we begin our discussion with a simple insightful model of $p$-electrons on a 2D honeycomb lattice[37] depicted in Fig. 2b. The tight-binding Hamiltonian assumes the form

$$H = \sum_{ij} t_{ij} c_i^\dagger c_j + \sum_i (\varepsilon_i \mathbb{1} + \mathbf{B} \cdot \boldsymbol{\sigma}) c_i^\dagger c_i + H_{\mathrm{soc}}, \tag{2}$$

where the first term is the hopping with $t_{ij}$ between orbitals $i, j = p_x, p_y, p_z$ on different sublattices, the orbital-dependent $\varepsilon_i$ is an on-site energy, the exchange field is $\mathbf{B} = B(\sin\theta, 0, \cos\theta)$, the SOC reads $H_{\mathrm{soc}} = \xi \mathbf{l} \cdot \boldsymbol{\sigma}$, and $\boldsymbol{\sigma}$ is the vector of Pauli matrices (see also Methods). While the reflection $\mathcal{M}$ with respect to the film plane is a symmetry of the planar lattice, buckling breaks this mirror symmetry.

We consider first the $\mathcal{M}$-broken model, known to be a quantum anomalous Hall insulator[37] over a wide range of model parameters. As exemplified in Fig. 2a for strong exchange, valence and conduction bands approach each other as the magnetization direction $\theta$ is tuned, which results in an emergent band crossing slightly off the $K$ point for $\theta \approx 60°$. Using our classification scheme, we identify this single mixed Weyl point as type-(ii) since it occurs for a generic magnetization direction, the value of which is controlled by the magnitude of SOC and exchange coupling. The effective Hamiltonian close to the linear crossing is governed by three tunable parameters, entangling momentum, magnetization direction, and the interactions of the model, which facilitates a degenerate point in the spectrum following the von Neumann–Wigner theorem[38]. Figure 2c

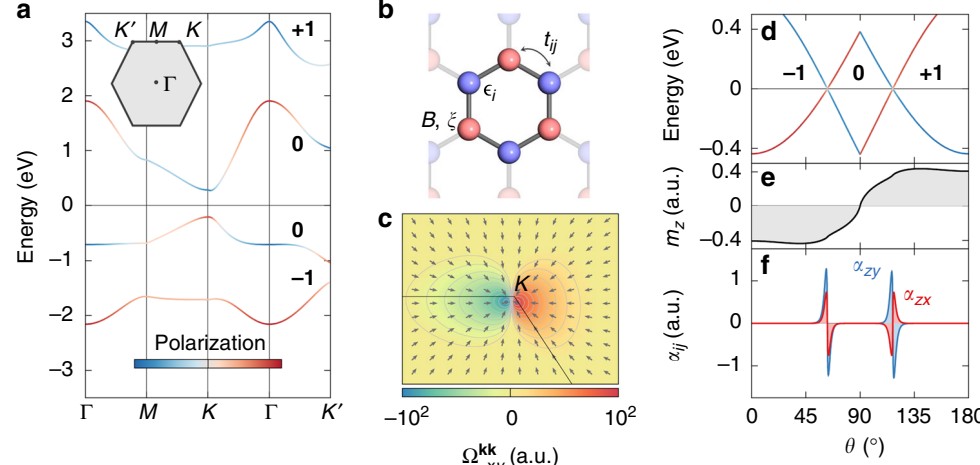

**Fig. 2** Model of a mixed topological semimetal. **a** Band structure for $\theta = 45°$, showing the lowest four energy bands of the $p$-model on the buckled honeycomb lattice. Bold numbers refer to the individual Chern numbers of the bands, and colors encode the states' polarization in terms of $p_x - ip_y$ (blue) and $p_x + ip_y$ (red) orbital character. **b** Honeycomb lattice of the model. (**c**) Distribution of the Berry curvature $\Omega_{xy}^{\mathbf{kk}}$ in momentum space close to the emergent nodal point for $\theta = 60°$. The in-plane direction of the full Berry curvature field $\mathbf{\Omega}$ is indicated by unit arrows that refer to the mixed curvatures $-\Omega_{yy}^{\widehat{\mathbf{mk}}}$ and $\Omega_{yx}^{\widehat{\mathbf{mk}}}$. **d–f** Evolution with respect to the magnetization direction $\theta$ of **d** the valence band top and conduction band bottom, **e** the total orbital magnetization $m_z$, and **f** the orbital Edelstein response $\alpha_{ij}$ using $k_BT = 25$ meV in the Fermi distribution, with the Fermi level set to the energy of the band crossing. In panel **d**, the Chern number $\mathcal{C}$ of the occupied states is bold, and colors denote the orbital polarization as in **a**

suggests that the isolated nodal point manifests in a characteristic distribution of the Berry curvature, whereby it mediates a topological phase transition from the non-trivial ($\mathcal{C} = -1$) to the trivial regime ($\mathcal{C} = 0$) as shown in Fig. 2d. The unique orbital signatures of such a mixed Weyl point, summarized in Fig. 2d–f, will be discussed later.

To uncover the non-trivial topology of the metallic point in the mixed space of Bloch vector and magnetization direction, we evaluate the flux of the Berry curvature field $\Omega$ through an enclosing surface in ($\mathbf{k}$, $\theta$)-space, which amounts to the topological charge $Q$ of that point according to Eq. (1). The mixed Weyl point emerging at $\theta \approx 60°$ carries a negative unit charge, which is consistent with the distribution of the Berry curvature field $\Omega$ in Fig. 2c. Figure 2d illustrates the presence of another nodal point located near $K'$ with the very same topological charge if the magnetization direction is tuned to $\theta \approx 120°$. However, the net topological charge over the full Brillouin zone of the combined phase space is zero since the two mixed Weyl points with negative unit charge are complemented by partners of opposite topological charge for $\theta \approx 240°$ and $\theta \approx 300°$, respectively. Owing to their topological protection, these generic mixed Weyl points feature a unique property: if the Hamiltonian is perturbed, they may only move to a different position in ($\mathbf{k}$, $\theta$)-space but cannot gap out easily. In the context of angle-resolved photoemission spectroscopy (ARPES) performed at a fixed magnetization direction, this means that although the generic nodal points might appear non-robust with respect to strain, chemical deposition of adsorbates, alloying etc., it should generally be possible to recover them again upon adjusting the magnetization direction. We anticipate that the topological charge can be measured by experiments that sweep the magnetization close to the nodal point: during the phase transition both the quantized anomalous Hall transport and the currents that are pumped by the magnetization dynamics, relating to the mixed Berry curvature[39], change uniquely as they are sensitive to the magnetic orientation of the ferromagnet.

While tuning the ratio between exchange and SOC expands or shrinks the extent of the topologically non-trivial phases, symmetries can enforce the appearance of the trivial state with $\mathcal{C} = 0$ under certain magnetization directions, resulting in a band crossing. To illustrate this, we turn to the planar honeycomb lattice that respects the mirror symmetry $\mathcal{M}$ with respect to the film plane. If the magnetization points along any in-plane direction, the combined symmetry of time-reversal and mirror operation requires that the Chern number vanishes. Therefore, starting from a non-trivial phase as induced by the model's interactions for finite out-of-plane magnetization, the system undergoes a topological phase transition from $\mathcal{C} = -1$ to $\mathcal{C} = 1$ exactly at $\theta = 90°$ (see Supplementary Fig. 1). Contrary to the buckled case, this transition is mediated by two nodal points located at $K$ and $K'$, respectively, each of which carries a negative unit topological charge. Thus, while the minimal number of type-(ii) nodal points is one for a given $\theta$, the symmetry-related type-(i) mixed Weyl points come at least in pairs of the very same charge. This correlates with a distinct nature of the topological phase transition in terms of the minimal change of the Chern number by $\Delta\mathcal{C} = \pm 1$ and $\Delta\mathcal{C} = \pm 2$, respectively. We anticipate that type-(i) mixed Weyl points may share formal analogies with symmetry-constrained counterparts[40] in 3D solids.

Remarkably, whereas adjacent nodal points in momentum space of conventional 3D Weyl semimetals must have opposite charges and thus annihilate under certain conditions[40] if pushed towards each other[11,12], this is not necessarily the case in 2D MWSMs. We speculate that it might be more difficult to destroy the nodal points in MWSMs by realistic perturbations of the Hamiltonian as they can have the same topological charge for a given magnetization direction, while their counterparts of opposite charge may be very far from them in $\theta$. Generally this does not imply that the direction under which the mixed Weyl points occur is constant as the Hamiltonian is perturbed, although this is the case in the planar model if the perturbation preserves symmetry. On the other hand, the two nodal points of negative charge, emerging originally at $\theta = 90°$, split into two distinct entities that manifest for generic directions of the magnetization if the restrictive symmetry is broken, e.g., due to buckling of the lattice.

**From topological insulators to mixed Weyl semimetals.** According to our model analysis, the interplay between magnetism and topology in 2D materials offers the potential to realize mixed Weyl points with non-zero topological charges. We apply electronic-structure methods to uncover these nodal points in first candidates of single-layer ferromagnets with SOC. As prototypical examples that are very susceptible to external magnetic fields, and display a rich topological phase diagram, we choose TlSe[41], Na$_3$Bi[42], and GaBi[43] (see Supplementary Note 1), which are originally TCIs and/or TIs with large energy gaps (for the unit cells see insets in Fig. 3). To study systematically the mixed topology, we use an additional exchange field term $\mathbf{B} \cdot \boldsymbol{\sigma}$ on top of the non-magnetic Hamiltonian.

We start by considering the case of planar TlSe, which is a TCI if no exchange field is applied[41]. Introducing an exchange field with an in-plane component breaks both time-reversal and $\mathcal{M}$-mirror symmetry, provides an exchange splitting between spin-up and spin-down states, and brings conduction and valence bands closer together. As follows from our topological analysis, the TCI character is kept even under sufficiently small exchange fields (see Supplementary Note 2). In analogy to the $\mathcal{T}$-broken quantum spin Hall insulator[44], we refer to this phase as a 2D $\mathcal{M}$-broken TCI. Increasing the magnitude $B$ results in a gap closure and gives rise to a non-trivial semimetallic state at the critical value $B_c$. If the exchange field exceeds this value, the reopening of the energy gap is accompanied by the emergence of the quantum anomalous Hall phase for any magnetization direction with finite out-of-plane component (see Fig. 3a and Supplementary Note 3).

Now, we turn to the in-plane magnetized system that exhibits symmetry, and for which the gap closes over a wide range of fields $B > B_c$, see Fig. 3a. As exemplified in Fig. 4a, the electronic structure for $B > B_c$ reveals that the gap closing is mediated by four isolated metallic points around the Fermi energy, where bands of opposite spin cross slightly off the $X$ and $Y$ points. Owing to their characteristic Berry curvature field, Fig. 4c, each of these mixed Weyl points occuring for $\theta = 90°$ carries a positive unit charge, which corresponds to a change of the Chern number $\mathcal{C}$ from $+2$ to $-2$. Consequently, the Berry phase $\gamma$ evaluated along a closed loop in momentum space around one of the points acquires a value of $\pi$. In total, the topological charge over the full phase space vanishes as the individual charges of the mixed Weyl points at $\theta = 90°$ are compensated by four nodal points that emerge during a second topological phase transition at $\theta = 270°$, Fig. 4c. Analogously to the planar model, we classify these objects as type-(i) nodal points since their emergence is enforced by the symmorphic symmetry, contrary to the Dirac nodes in 2D Dirac semimetals that are protected by non-symmorphic symmetries[20]. The non-trivial mixed topology further leads to exotic boundary solutions in finite ribbons of TlSe, Fig. 4b.

Breaking the underlying $\mathcal{T} \otimes \mathcal{M}$ symmetry, e.g., by buckling of the lattice (see Supplementary Note 2), splits the four nodal points in TlSe, which originally appeared at $\theta = 90°$, into two distinct groups that manifest for generic magnetization directions.

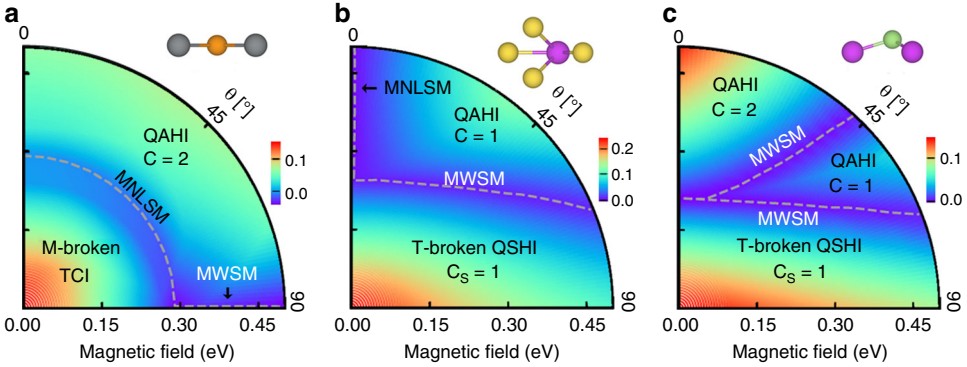

**Fig. 3** Emergence of mixed topological semimetallic states. Phase diagrams of one-layer **a** TlSe, **b** Na$_3$Bi, and **c** GaBi with respect to the magnitude $B$ and the direction $\hat{\mathbf{m}} = (\sin\theta, 0, \cos\theta)$ of the applied exchange field. Side views of the unit cells highlight differences in the crystalline symmetries, and colors represent the value of the global band gap in eV. To characterize quantum spin Hall (QSHI) and quantum anomalous Hall (QAHI) phases, we use the spin Chern number $\mathcal{C}_S$ and the momentum Chern number $\mathcal{C}$, respectively. Dashed gray lines mark the boundary between different insulating topological phases for which the band gap closes. The emergent metallic states are labeled as either mixed Weyl semimetal (MWSM) or mixed nodal-line semimetal (MNLSM)

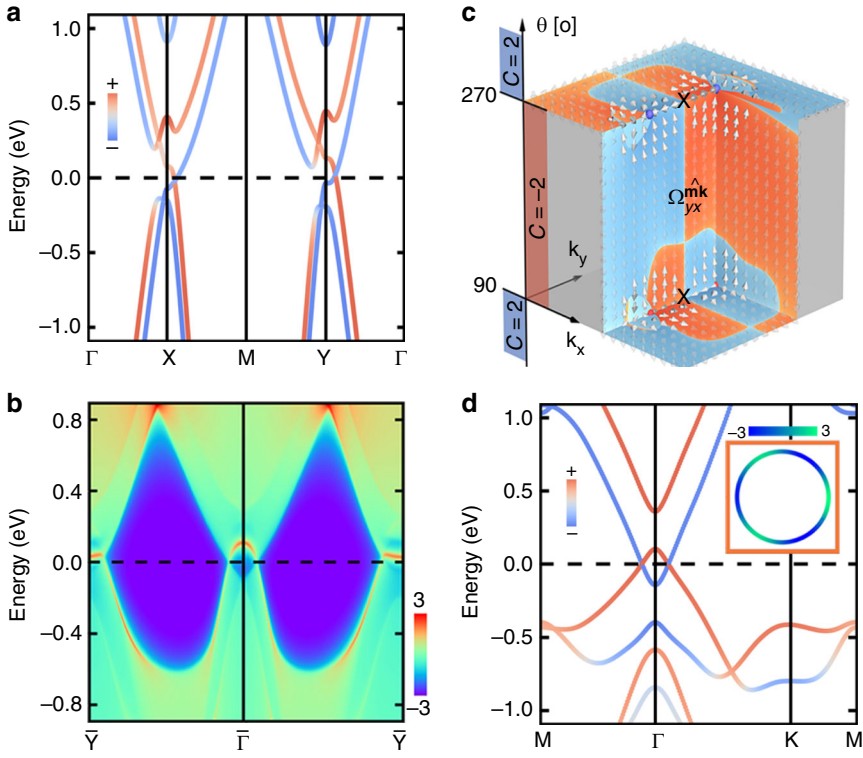

**Fig. 4** Electronic properties of mixed topological semimetals. **a** Spin-resolved band structure and (**b**) energy dispersion of a finite ribbon, where the localization at the edge is indicated by colors ranging from blue (weak) to red (strong), of the TlSe monolayer with an in-plane exchange field of magnitude $B = 0.5$ eV. **c** Around the $X$ point, for example, the emergence of nodal points with opposite topological charge (red and blue balls) for reversed in-plane directions $\theta$ of the magnetization imprints characteristic features on the phase-space distribution of the Berry curvature field $\Omega = (-\Omega_{yy}^{\widehat{\mathbf{mk}}}, \Omega_{yx}^{\widehat{\mathbf{mk}}}, \Omega_{xy}^{\mathbf{kk}})$ as indicated by the arrows. A logarithmic color scale from blue (negative) to red (positive) is used to illustrate the mixed Berry curvature $\Omega_{yx}^{\widehat{\mathbf{mk}}}$ in the complex phase space of $\mathbf{k}$ and $\theta$. **d** Spin-resolved band structure of one-layer Na$_3$Bi with an exchange field of magnitude $B = 0.5$ eV applied perpendicular to the film. Owing to the mirror symmetry of the system, the band crossings around $\Gamma$ form a mixed nodal line in momentum space, which disperses as illustrated in the inset, where colors indicate energy differences between the crossing points and the Fermi level in meV

To elucidate this transition more clearly, we consider the monolayers Na$_3$Bi and GaBi, where this symmetry is absent. As visible from the phase diagrams in Fig. 3b, c, the single Weyl points in these systems emerge at the boundaries between the $\mathcal{T}$-broken quantum spin Hall phase and Chern insulator phases with different Chern numbers. Analogously to TlSe, we identify the mixed topological charge of such points to be $Q = \pm 1$, depending on the position in $(\mathbf{k}, \theta)$-space. However, in contrast to

TlSe, for which both the number as well as the position of mixed Weyl points is determined by symmetry, the single mixed Weyl point in Na$_3$Bi and GaBi appears for a given generic direction of the magnetization, and we thus classify it as type-(ii) nodal point.

**From mixed Weyl points to mixed nodal lines.** It can occur that the mixed Weyl point is realized accidentally for a range of $\theta$ in

the 2D ferromagnet, as it is exemplified in TlSe at a fixed value of exchange field of about 0.29 eV, see Fig. 3a. In the spirit of Fig. 1d, this presents a truly mixed nodal line, a 1D manifold of states, which evolves not only in $k$-space but also in $\theta$. The topological character of the line is reflected in the Berry phase that is the line integral of the Berry connection along a path in $k$-space, which encloses the corresponding point that the mixed nodal line pinches in the Brillouin zone at a given $\theta$, see Fig. 1d. The occurrence of such mixed nodal lines is purely accidental and does not rely on symmetries, while perturbing the system (i.e., by changing the magnitude of **B**) may result in the mixed nodal line's destruction, as we discuss below. Owing to the subtle interplay of exchange interactions and relativistic effects which underlies their emergence, the realization of such mixed nodal lines in real materials sets an exciting challenge, where 2D magnets are advantageous for semimetallic states that are robust against variations of the magnetization direction.

Another distinct type of a mixed nodal line is the 1D nodal line, which evolves in $k$-space for a fixed direction of the magnetization, see Fig. 1c, similarly to the nodal-line semimetals which exhibit nodal lines in high-symmetry planes corresponding to the crystalline mirror symmetry[13–15]. While the $\mathcal{M}$ symmetry is broken by an in-plane exchange field, it survives when $\hat{\mathbf{m}}$ is perpendicular to the film. As shown in Fig. 3b, the energy gap remains closed in Na$_3$Bi with $\theta = 0°$ above the critical magnitude $B_c$. To gain insights into the topological properties in this case, we take $B = 0.5$ eV and present the spin-resolved band structure of the system in Fig. 4d. In absence of inversion and time-reversal symmetries, all bands are generically non-degenerate. Taking into account the mirror symmetry $\mathcal{M}$, bands in the $xy$-plane can be marked by mirror eigenvalues $\pm i$, and those with opposite mirror eigenvalues can cross each other without opening a gap. As the highest occupied and lowest unoccupied bands in Na$_3$Bi cross each other around the $\Gamma$ point, a nodal line is formed as shown in the inset of Fig. 4d.

To validate the mixed topological character of this nodal line in Na$_3$Bi, we compute its non-trivial Berry phase according to Fig. 1c. However, the overall flux of the generalized Berry curvature field through any surface in ($\mathbf{k}$, $\theta$)-space surrounding the mixed nodal line vanishes, which confirms the zero topological charge of the mixed nodal line as an object in 3D space of momentum and $\hat{\mathbf{m}}$. Accordingly, there is no variation in the Chern number $\mathcal{C} = +1$ as the magnetization crosses $\theta = 0°$ (see Supplementary Fig. 5b), which signifies the lack of topological protection of the mixed nodal line and its disappearance as the mirror symmetry is broken upon turning $\hat{\mathbf{m}}$ away from the $z$-axis. To prove that the nodal line originates from the mirror $\mathcal{M}$, we perform for $\theta = 0°$ various distortions of the lattice that break the three-fold rotational axis perpendicular to the film but preserve $\mathcal{M}$, resulting still in the mixed nodal line though for different strengths of the exchange field (see Supplementary Fig. 7).

**Mixed topological semimetals in feasible 2D ferromagnets.** Having established the existence of mixed topological semimetals in a simple model and by applying an external exchange field to TIs/TCIs, an important question to ask is whether the proposed mixed topological semimetals can be realized in stable 2D ferromagnets. While the existence of mixed Weyl points in several 2D magnets such as doped graphene or semi-hydrogenated bismuth has been shown[30], in this work we demonstrate the possibility of their realization in other realistic systems, aiming especially at van der Waals crystals. Bulk VOI$_2$ has a layered structure characterized by the orthorhombic space group $Immm$, and has already been synthesized and investigated[45,46]. We focus

on a VOI$_2$ monolayer, the unit cell of which contains two I, one O, and one V atom that is coordinated in the center as shown in Fig. 5a. The electronic structure of the single layer represents the in-plane electronic structure of its bulk parent compound quite well, and one-layer fabrication could be realized experimentally, e.g., by mechanical exfoliation from the layered bulk due to the low cleavage energy of 0.7 meV/Å$^2$, which is much smaller than for graphite (12 meV/Å$^2$) or MoS$_2$ (26 meV/Å$^2$). As verified by our explicit calculations of the phonon spectrum, the monolayer is dynamically stable and difficult to destroy once formed. The ground state of the system is ferromagnetic with a spin magnetization of about 1 $\mu_B$ per unit cell and an easy in-plane anisotropy. Supplementary Note 4 presents further details on the electronic structure.

As illustrated in Fig. 5b, the band structure of one-layer VOI$_2$ with in-plane magnetization and SOC reveals band crossings along the $M - Y$ and $\Gamma - X$ paths near the Fermi level. There are four semimetallic points in the 2D Brillouin zone as can be seen from the $k$-resolved energy difference between top valence band and lowest conduction band around $\Gamma$ and Y (see Fig. 5b). To demonstrate the topological nature of these points we analyze the distribution of the Berry curvature in the complex phase space shown in Fig. 5b, and find that all four crossings are mixed Weyl points with a charge of $+1$. Similarly to magnetized TlSe discussed before, the predicted mixed Weyl points in VOI$_2$ monolayer are protected by the operation as can be confirmed explicitly by breaking this symmetry, which gaps out the nodal points. When constructing a semi-infinite 1D ribbon of the material along the $\Gamma - Y$ direction, we observe that the four mixed Weyl points project onto two pairs of distinct points that are connected by emergent edge states close to $\bar{X}$ and $\bar{\Gamma}$ as illustrated by the edge dispersion in Fig. 5c.

In addition, to prove the emergence of mixed nodal lines in realistic 2D magnets, we start from Na$_3$Bi in its hexagonal $P6_3/mmc$ phase, which is one of the first established realizations of Dirac semimetals. This material has been synthesized both in bulk and film form[8,9,47]. Replacing one Na atom with Cr, we focus here on a monolayer of Na$_2$CrBi (see Fig. 5d for a sketch of the unit cell), which is a strong ferromagnet that is energetically and dynamically stable according to our calculations of cleavage energy (25.5 meV/Å$^2$) and phonon spectrum (see Supplementary Fig. 8). Including SOC, the band structure of perpendicularly magnetized Na$_2$CrBi in Fig. 5d exhibits prominent band crossings around the $\Gamma$ point. These metallic points form due to the mirror symmetry $\mathcal{M}$, which allows two bands with opposite mirror eigenvalues to cross close to the Fermi level. Extending the analysis to the full Brillouin zone reveals that these band crossings forge a nodal loop (see Fig. 5e). In analogy to the previous case of the Na$_3$Bi monolayer, the nodal line is gapped out as soon as the mirror symmetry is broken, e.g., by tilting the magnetization direction, and we verify its non-trivial mixed topology by evaluating the Berry phase around the nodal line as outlined in Fig. 1c. As in the case of VOI$_2$, a key manifestation of the complex topology of the mixed nodal line is the emergence of characteristic edge states in a semi-infinite ribbon of Na$_2$CrBi, which can be clearly distinguished from the projected bulk states in Fig. 5f.

**Origin of mixed nodal points and orbital magnetism.** Finally, we elucidate one of the universal physical mechanisms that triggers magnetically induced topological phase transitions and gives rise to non-trivial band crossings. We refer again to the above model, which contains elementary ingredients that govern the appearance of mixed nodal points, including exchange and SOC (see Eq. (2)). As illustrated in Fig. 2d, given an initial spin-orbit driven energy splitting for out-of-plane magnetization,

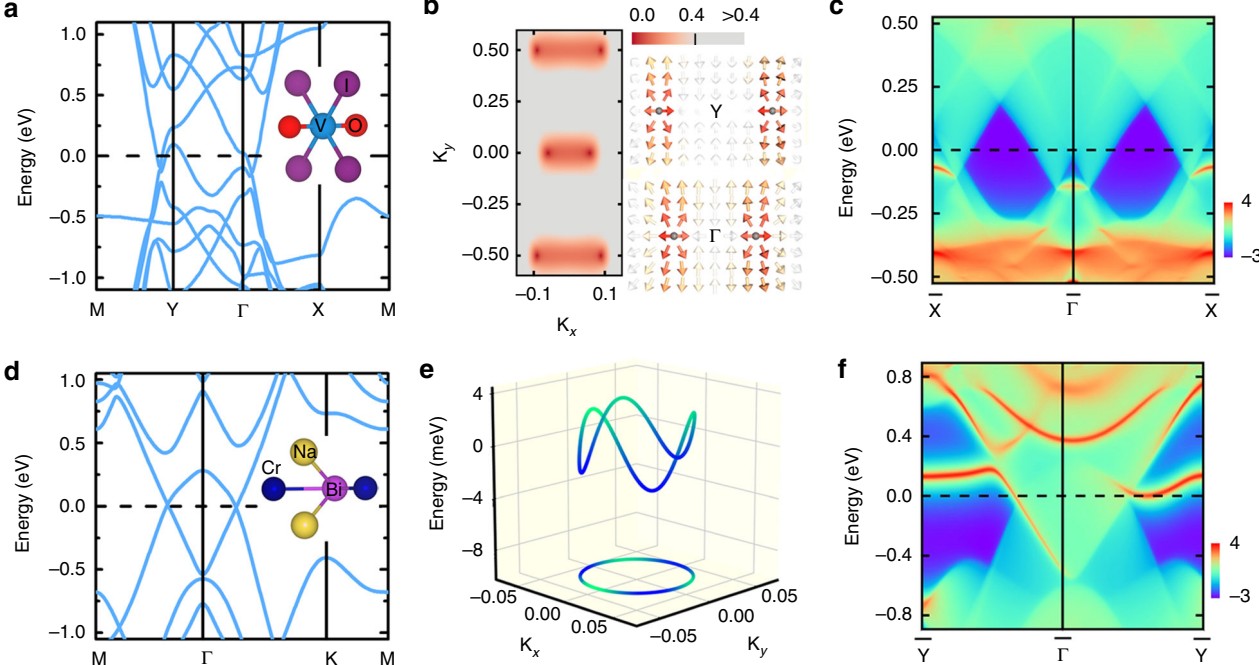

**Fig. 5** Realization of mixed topological semimetals in two-dimensional ferromagnets. Including SOC, the electronic band structures of the stable single-layer compounds **a** $VOI_2$ and **d** $Na_2CrBi$ display band crossings with non-trivial topological properties in the mixed phase space of crystal momentum $\mathbf{k} = (k_x, k_y)$ and magnetization direction $\theta$. Side views of the corresponding unit cells are shown as insets. **b** The signatures of the mixed Weyl points for in-plane magnetized $VOI_2$ manifest in the momentum-resolved direct band gap (color scale in eV) and in the distribution of the Berry curvature field shown as arrows throughout the complex phase space, confirming the presence of four nodal points with the same topological charge. Ranging from white (small) to dark red (large), the arrow's color illustrates the magnitude of the Berry curvature field. **c** Electronic structure of a semi-infinite ribbon of $VOI_2$, where the state's localization at the edge is indicated by colors ranging from dark blue (weak) to dark red (strong). **e** In the mirror-symmetric plane, the band crossings in perpendicularly magnetized $Na_2CrBi$ form a mixed nodal line that disperses in energy. **f** The band structure of a semi-infinite $Na_2CrBi$ ribbon reveals characteristic edge states due to the non-trivial mixed topology

bands with different orbital character are guaranteed to cross as the direction of the magnetization is reversed, owing to the fact that the orbital momentum of the system is dragged by the magnetization via SOC. This observation is well known in molecular physics as well as in the band theory of ferromagnets[48]. In the studied model such crossing points of orbitally polarized states appear either for $\theta = 90°$ enforced by the symmetry, or under generic directions of the magnetization if the symmetry in the orbitally complex system is reduced. Although the non-trivial mixed topology originates here primarily from $p$-states, we point out that materials with $s$-electrons on a bipartite lattice can offer similar prospects by exploiting the valley degree of freedom[30].

Since the inversion of the orbital chemistry mediates the level crossing, we argue that this transition imprints general magnetic properties. A representative example of a real material where the crossing emerges at general $\theta$ is the semi-hydrogenated bismuth film H-Bi, which has been shown to host single mixed Weyl points at $\theta \approx \pm 43°$ and $\pm 137°$ due to a magnetically induced topological phase transition from a Chern insulator with $\mathcal{C} = \pm 1$ to a trivial phase[30]. In Fig. 6a we plot the evolution of the orbitally resolved electronic structure of H-Bi around the Fermi energy, where the states originate mainly from $p$ orbitals of bismuth. In accordance with the model scenario, the emergent nodal point correlates with a reordering of the $p_x \pm ip_y$ states, which underlines the role of SOC in mediating the inversion of energy bands in terms of their orbital character and stabilizing generic mixed nodal points.

The changes in the orbital character of the states across mixed Weyl points manifest in prominent changes in the local orbital magnetization (OM) near the mixed Weyl points as the

magnetization is varied. According to its modern theory[49–52], the OM as a genuine bulk property of the ground-state wave functions $|u_{\mathbf{k}n}^\theta\rangle$ is given by $m = \int \mathbf{m}(\mathbf{k})d\mathbf{k}$, with momentum-resolved contributions from all occupied bands

$$\mathbf{m}(\mathbf{k}) = \frac{e}{2\hbar} \operatorname{Im} \sum_n^{\mathrm{occ}} \langle \partial_\mathbf{k} u_{\mathbf{k}n}^\theta | \times [H_\mathbf{k} + \mathcal{E}_{\mathbf{k}n} - 2\mathcal{E}_\mathrm{F}] | \partial_\mathbf{k} u_{\mathbf{k}n}^\theta \rangle, \quad (3)$$

with $H_\mathbf{k} = e^{-i\mathbf{k}\cdot\mathbf{r}} H e^{i\mathbf{k}\cdot\mathbf{r}}$ as the lattice-periodic Hamiltonian, $\mathcal{E}_{\mathbf{k}n}$ as the energy of band $n$, and $\mathcal{E}_\mathrm{F}$ as the Fermi level.

Equation (3) underlines the deep relation of the OM to the local geometry in $k$-space, and it is thus expected that, in accord with the strongly modified geometry of Bloch states in $(\mathbf{k}, \theta)$-space in the vicinity of mixed Weyl points, the OM may also experience a pronounced variation both in $\mathbf{k}$ and $\theta$. Therefore, we anticipate that the non-trivial topology of the mixed Weyl points enhance the variation of the orbital character of the states. Indeed, our calculations verify the validity of this line of thought: Fig. 6d, e reflects unique local fingerprints and colossal magnitude of the orbital magnetization $m_z(\mathbf{k})$ in momentum space, which correlate with the emergence of magnetic monopoles in two of the predicted mixed topological semimetals. These features are present for both types of nodal points, i.e., the generic and symmetry-enforced ones.

Remarkably, the pronounced but competing local contributions to the OM for fixed $\theta$ nearly cancel each other, rendering the net effect of the mixed Weyl points on the total OM rather small. However, the microscopic response of the orbital chemistry to magnetically controlled band crossings opens up bright avenues for generating large orbital magnetization by applying

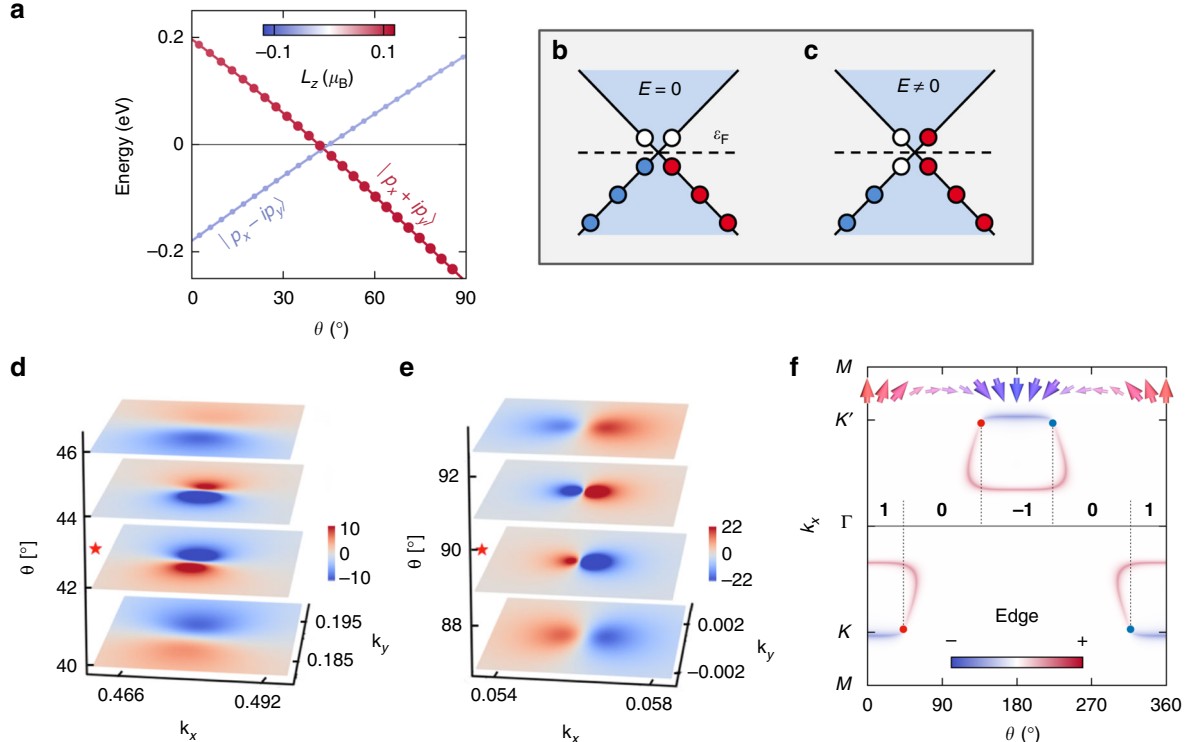

**Fig. 6** Microscopics and prospects of nodal points in mixed topological semimetals. **a** The $p$-dominated valence and conduction states in the functionalized bismuth film realize an orbital inversion close to the Fermi energy, leading to an emergent band crossing for the generic direction $\theta = 43°$. The $z$-component of the orbital angular momentum $\mathbf{L} = -\mu_B \sum_{\mathbf{k}n}^{occ} \sum_{\mu} \langle \psi_{\mathbf{k}n}^{\theta} | \mathbf{r}^{\mu} \times \mathbf{k} | \psi_{\mathbf{k}n}^{\theta} \rangle_{\mu}$ of all occupied Bloch states $|\psi_{\mathbf{k}n}^{\theta}\rangle$ is represented by colors, $\mathbf{r}^{\mu}$ is the position relative to the $\mu$th atom, and the real-space integration is restricted to spherical regions around the atoms. **b, c** A finite electric field $\mathbf{E}$ repopulates the electronic states at the Fermi level $\mathcal{E}_F$, which can be used to promote the net effect of mixed Weyl points on orbital magnetism. **d, e** Evolution of the orbital magnetization $m_z(\mathbf{k})$ in the complex phase space of the crystal momentum $\mathbf{k}$ and $\theta$ in **d** the functionalized bismuth bilayer, and **e** the ferromagnet VOI$_2$. In both cases, the topological phase transition, which is accompanied by an emergent monopole in momentum space, happens for the critical value of $\theta$ that is indicated by the red star. **f** One-dimensional Fermi arcs connect the projections of the nodal points with opposite charge (red and blue dots) in a zigzag ribbon of the functionalized bismuth bilayer. Red and blue colors denote the localization of the Fermi arcs at opposite edges, and bold numbers refer to the evolution of the bulk Chern number $\mathcal{C}$ with the magnetization direction $\theta$

an electric field that repopulates the occupied states (see Fig. 6b, c). Such a giant current-induced orbital Edelstein effect can have a strong impact on phenomena that rely sensitively on the orbital moment at the Fermi surface. Moreover, the drastic change in the local OM with $\theta$ may be used to detect experimentally the presence of mixed Weyl points in the electronic structure by detecting large variations in the current-induced orbital properties[53,54]. To demonstrate the feasibility of our proposal, we evaluate the orbital Edelstein effect $m_i = \alpha_{ij} E_j$ for the buckled $p$-model with broken inversion symmetry within a Boltzmann theory:[53,54]

$$\alpha_{ij} = e\tau \sum_n \int \frac{d\mathbf{k}}{(2\pi)^2} \frac{df}{d\mathcal{E}_{\mathbf{k}n}} m_{n,i}^{loc}(\mathbf{k}) v_{n,j}(\mathbf{k}), \qquad (4)$$

where $\tau$ is the relaxation time, $f$ is the Fermi distribution function, and $v_{n,i}(\mathbf{k})$ and $m_{n,i}^{loc}(\mathbf{k})$ correspond to the $i$th components of the state's group velocity and its local orbital moment $\mathbf{m}_n^{loc}(\mathbf{k}) = (e/2\hbar) \text{Im} \langle \partial_{\mathbf{k}} u_{\mathbf{k}n}^{\theta} | \times [H_{\mathbf{k}} - \mathcal{E}_{\mathbf{k}n}] | \partial_{\mathbf{k}} u_{\mathbf{k}n}^{\theta} \rangle$, respectively. While the equilibrium OM hardly changes as a function of the direction $\theta$ (see Fig. 2e), the sharply peaked current-induced response $\alpha_{ij}$ is an immediate orbital signature of the emergent mixed Weyl points with complex topology, Fig. 2f.

## Discussion

Owing to the nature of mixed semimetals incorporating the magnetization direction as an integral variable, we expect pronounced

topological magneto-electric effects to which these materials should give rise. Apart from their substantial relevance for technological applications based on magnetic solids, we anticipate that these coupling phenomena can play a key role even in finite systems such as quantum dots[55]. Analogously, we envisage that complementing the topological classification of matter by magnetic information rooting in the electronic degrees of freedom will be valuable for other research fields as well, e.g., for topological magnon semimetals[56,57]. In addition to the prospects for current-induced orbital magnetism, current-induced spin-orbit torques that can be used to efficiently realize topological phase transitions, and possible giant influences on anisotropic magnetotransport[30,31,58], we would like to emphasize in particular the promises of mixed semimetals for chiral magnetism. While it is known that MWSMs may exhibit a distinct tendency towards chiral magnetism[30], we speculate that chiral spin textures such as magnetic skyrmions or domain walls can effectively unravel the topological features of mixed semimetals in real space, which can have profound consequences on, e.g., orbital magnetism and transport properties of these textures.

To illustrate this point more clearly, we return to the emergent nodal points in H-Bi. In complete analogy to 3D Weyl semimetals in momentum space, the complex mixed topology of MWSMs results generally in the emergence of mixed Fermi arcs at the surfaces of these systems. By following the $\theta$-evolution of the electronic structure of 1D ribbons of H-Bi that are periodic along the $x$-axis, we effectively construct such a 2D surface for which we

present in Fig. 6f the states at the Fermi energy as a function of $k_x$ and $\theta$. As clearly evident, the emergent surface states connect the projections of the mixed Weyl points with opposite charge, realizing mixed Fermi arcs. Imagining now a long-wavelength chiral domain-wall running along the $x$-axis of a H-Bi ribbon, where $\theta(x)$ describes the local variation of the magnetization, we recognize that the mixed Fermi arcs will manifest in topological metallic states in certain regions of the domain-wall as a consequence of the non-trivial mixed topology. In chiral spin textures hosting mixed Weyl semimetallic states, we anticipate that such electronic puddles will result in topologically distinct contributions to the current-driven spin torques acting on these spin structures, a prominent variation of the texture-induced Hall signal in real space, chiral and topological orbital magnetism[59–61], as well as topological contributions to the longitudinal transport properties of domain walls and chiral magnetic skyrmions made of MWSMs. Interfaces between topological insulators and dynamic magnetization structures present further compelling examples of such complex mixed topologies[62–64]. We thereby proclaim that exploring the avenues associated with the exotic electronic, transport, and response phenomena in textured mixed semimetals presents one of the most exciting challenges in topological chiral spintronics.

## Methods

**Tight-binding model**. In order to arrive at the model (2) of the mixed Weyl semimetal, we extended the tight-binding Hamiltonian of ref. [37] to describe arbitrary magnetization directions. Throughout this work we incorporated only the nearest-neighbor hopping of $\sigma$-type on the honeycomb lattice with $t_\sigma = 1.854$ eV but our general conclusions remain valid even beyond nearest-neighbor hoppings. In addition, we chose $B = 8.0$ eV to fully spin-polarize the bands, $\xi = 1.0$ eV for the SOC, and we shifted the $p_z$ states to higher energies. By introducing a relative shift of the on-site energies we further imitated the buckling of the honeycomb lattice. Diagonalizing at every $(\mathbf{k}, \theta)$-point the Fourier transform of the $12 \times 12$ matrix that results from Eq. (2) grants access to the wave functions and the band energies.

**First-principles calculations**. Based on density functional theory as implemented in the full-potential linearized augmented-plane-wave code FLEUR (see http://www.flapw.de), we converged the electronic structure of the studied systems including SOC self-consistently. Exchange and correlation effects were treated in the generalized gradient approximation of the PBE functional[65]. To represent the electronic Hamiltonian efficiently, we subsequently constructed so-called maximally localized Wannier functions using the wannier90 program[66,67]. In this tight-binding basis, the Hamiltonian of the non-magnetic TI/TCI systems was supplemented by an exchange term $\mathbf{B} \cdot \sigma$, where $\sigma$ is the vector of Pauli matrices and $\mathbf{B} = B(\sin\theta, 0, \cos\theta)$. In the ferromagnetic candidate materials, we obtained an efficient description of the electronic structure in the complex phase space of $\mathbf{k}$ and $\theta$ by constructing a single set of higher-dimensional Wannier functions[68]. The structural relaxations of the ferromagnetic systems were carried out in the Vienna Ab initio Simulation Package[69].

## Data availability

The tight-binding code and the data that support the findings of this study are available from the corresponding authors on reasonable request.

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

## Acknowledgements

This work was supported by the Priority Program 1666 of the Deutsche Forschungsgemeinschaft (DFG), the Virtual Institute for Topological Insulators (VITI), the Natural Science Foundation of Shandong Province under Grant No. ZR2019QA019, and the Qilu Young Scholar Program of Shandong University. This work has been also supported by the Deutsche Forschungsgemeinschaft (DFG) through the Collaborative Research Center SFB 1238. We acknowledge computing time on the supercomputers JUQUEEN and JURECA at Jülich Supercomputing Centre and JARA-HPC of RWTH Aachen University.

## Author contributions

C.N. and J.-P.H. performed the first-principles calculations and model analysis. Y.M. with contributions from C.N., J.-P.H. and G.B. conceived the concept and designed the research. C.N., J.-P.H. and Y.M. wrote the manuscript with contributions from P.M.B., H.Z., L.P., D.W., G.B. and S.B.

## Additional information

**Competing interests:** The authors declare no competing interests.

