## [Peer Review File · Nature Communications]

Reviewers' comments:

Reviewer #1 (Remarks to the Author):

This paper proposes the possible topological semimetal phase, including Weyl semimetal and nodal-line semimetal phases, in two dimensional ferromagnetic materials in the mixed space of 2D momentum space and magnetization direction. The idea of classifying topological semimetal phases in certain mixed space (instead of pure momentum space) itself is quite interesting and may have some important impact in the field of topological semimetal. However, I do not think the current manuscript is easy to read and has achieved the high standard of Nature Communications.

1. For the section of the discussion of three materials TlSe, Na₃Bi and GaBi, it is clear that the author is not to propose realistic system, since none of them are magnetic materials and cannot induce a strong exchange type of coupling of 0.1eV. However, as a model study, it is quite difficult to figure out the underlying physics from such complex model system. In my mind, some minimal model study is much more illustrative for the purpose of this section. For example, in the study of TlSe, the author claims the gapless nodes are stabilized by T*M symmetry. But it is unclear how the T*M symmetry stabilizes the gapless nodes in 2D theoretically. I think some type of effective model will be helpful to this end. For example, the authors can construct the effective model around the gapless nodes and try to show how T*M symmetry acts on this effective model and why the nodes are stable.

More generally, I think the authors should give some theoretical argument about the condition for topological semimetal phase in 2D ferromagnetic materials. What type of crystalline symmetry is required? What types of atomic orbitals are required?

The author claims two mechanisms for such mixed Weyl points, but for the (ii) condition, it is unclear to me the meaning of "complex interplay of exchange interaction and spin-orbit coupling in systems of low symmetry." I think the authors should specify this mechanism. For "generic band crossing", I think the authors should consider von Neumann-Wigner theorem of level's repulsion. I believe the effective model study will be useful in this context.

2. Two materials VOI₂ and Na₂CrBi are interesting, but the feasibility of these materials is essential. For VOI₂, it seems that the ref 43 only includes VOI₃, not VOI₂. For Na₂CrBi, I wonder if any experimental efforts in growing similar materials.

3. The experimental proposal about the possibility of giant current-induced orbital Edelstein effect is also interesting. However, the current work only shows the distribution of orbital magnetization in the momentum space. I think a direct calculation of the giant current-induced orbital Edelstein effect based on Ref. 48 and 49 is required to illustrate the experimental feasibility.

4. A small problem:

In page 7, there is a statement "This observation is well known in molecular physics as well as in band theory of ferromagnets." I think a citation is required for this statement.

Reviewer #2 (Remarks to the Author):

Authors study metallic states occurring at transitions between different topological phases induced by varying the strength and orientation of exchange fields induced by magnetization in two dimensional magnets. When viewed in the combined momentum and exchange field orientation space, such states appear as nodes and/or nodal lines, which carry non trivial Berry phases. In close analogy to appearance of such metallic states in pure momentum space (so-called Weyl semimetals) the states are termed here as "mixed Weyl semimetals".

In an earlier work, authors have already pointed out the existence of the mixed Weyl semimetal phase and its role in producing enhanced magneto-electric effects. The main contributions of the present

work are: (i) revealing example conditions under which mixed Weyl states arise starting from a topological crystalline/topological insulator phase (ii) finding natural material candidates harboring mixed Weyl semimetals, and (iii) possible manifestations of mixed Weyl semimetals, which include reordering of local orbital magnetization, presence of mixed fermi arcs and additional topological mode-induced torques in chiral magnetic configurations. These findings are supported by reasonable explanations which makes physical sense.

In recent years, there has been quite a bit of interest in the condensed matter community to find nontrivial manifestations of topology for fundamental and technological applications and I thus believe the present study is timely and of interest for Nature communication.

I have few questions and comments before I can recommend publication:

1. It is well known that metallic states arise at the boundary of topological phases induced by exchange fields and have been measured experimentally (see for example Science 358, 1311 (2017), and references therein). A major point of the present work is that these states when viewed in the mixed phase space carry a nontrivial topological charge. Although authors present possible indirect measurements of semi metallic nature (like ARPES as a function of magnetization and XMCD, see 2 below for XMCD related question), it will be nice if the authors could present an experimental scheme to directly measure this topological charge ?

2. Authors point out that the mixed Weyl nodes arise due to two mechanisms (i) presence of combined time reversal and mirror symmetries, and (ii) interplay of spin-orbit and exchange interactions. Later, they point out in a model system how spin-orbit interaction leads to mixed Weyl points, which are accompanied by a rearrangement of orbital character. Please clarify if the rearrangement of local orbital character is specific to spin-orbit coupling induced mixed Weyl nodes or is universal, i.e. it occurs for both mechanisms (i) and (ii)? This is specially of relevance for the proposed experimental observation of current-induced change in local orbital magnetization, as a signature of mixed Weyl node.

3. The authors show that in "textured magnetic Weyl semimetals" metallic puddles are formed. Additionally they speculate that such electronic puddles are expected to give rise to topological contributions to torques and Hall effects. Similar metallic puddles are known to form in a simple example of a magnetic domain wall exchange coupled to topological insulators (see for example PRL 108, 187201 (2012) and quantum anomalous Hall insulators [see for example PRB 92, 085416 (2015); PRB 94, 020411(R) (2016)], where topological metallic puddles-induced domain wall motion has also been studied. Naively, it looks like these studied systems are an instance of "textured magnetic Weyl semimetals" proposed by the authors. Can authors comment on this or compare their case with the above mentioned studies ?

4. On a related note to point 3, according to earlier work by the authors [Nature Comm. 8, 1479 (2017)], even outside the Weyl semimetal phase large non-dissipative torques can appear on application of electric fields. It is not obvious how topological puddles-induced torques can be separated from such contributions. Can authors clarify this point?

We would like to thank the two reviewers for their critical reading as well as for providing valuable remarks that helped us to significantly improve our manuscript. In the following, we present our point-by-point response to all raised issues.

Overview of changes based on reviewers' comments:

- Introduced a perceptive model to illustrate physics of mixed Weyl points
- Included explicit calculations of orbital Edelstein effect
- Introduced new Fig. 2 to summarize results of the model
- Clarified the role of symmetry and orbital chemistry
- Clarified the distinction between type-(i) and type-(ii) nodal points
- Proposed an experimental scheme to measure topological charge
- Included the suggested references
- Condensed the paragraphs on the magnetized topological insulator materials
- Slightly modified the title of the manuscript
- New paragraph in the introduction emphasizes that the unique interplay of topology with magnetism is an important emerging research field

Reviewer #1: This paper proposes the possible topological semimetal phase, including Weyl semimetal and nodal-line semimetal phases, in two dimensional ferromagnetic materials in the mixed space of 2D momentum space and magnetization direction. The idea of classifying topological semimetal phases in certain mixed space (instead of pure momentum space) itself is quite interesting and may have some important impact in the field of topological semimetal. However, I do not think the current manuscript is easy to read and has achieved the high standard of Nature Communications.

Reply: We thank the reviewer for considering the proposed idea of topological semimetals in mixed phase spaces as *“quite interesting”* and that it *“may have some important impact in the field of topological semimetal”*. We hope that we have **enhanced readability of the manuscript** by considering the reviewer’s insightful suggestions, and that the reviewer considers our revised manuscript suitable for publication in Nature Communications.

Reviewer #1: For the section of the discussion of three materials TlSe, Na₃Bi and GaBi, it is clear that the author is not to propose realistic system, since none of them are magnetic materials and cannot induce a strong exchange type of coupling of 0.1eV. However, as a model study, it is quite difficult to figure out the underlying physics from such complex model system. In my mind, some minimal model study is much more illustrative for the purpose of this section. For example, in the study of TlSe, the author claims the gapless nodes are stabilized by T*M symmetry. But it is unclear how the T*M symmetry stabilizes the gapless nodes in 2D theoretically. I think some type of effective model will be helpful to this end. For example, the authors can construct the effective model around the gapless nodes and try to show how T*M symmetry acts on this effective model and why the nodes are stable.

More generally, I think the authors should give some theoretical argument about the condition for topological semimetal phase in 2D ferromagnetic materials. What type of crystalline symmetry is required? What types of atomic orbitals are required? The author claims two mechanisms for such mixed Weyl points, but for the (ii) condition, it is unclear to me the meaning of *“complex interplay of exchange interaction and spin-orbit coupling in systems of low symmetry.”* I think the authors should specify this mechanism. For *“generic band crossing”*, I think the authors should consider von

Neumann-Wigner theorem of level's repulsion. I believe the effective model study will be useful in this context.

Reply: We thank the reviewer for these insightful suggestions that we followed to restructure our manuscript, which clarifies now the physical origins for the emergence of nodal points in the mixed parameter space. Specifically, the revised manuscript starts from a simple but perceptive model of p-electrons on the honeycomb lattice, which is either buckled or planar. As illustrated in Fig. 2 and Supplementary Figure 1, this model exhibits topological phase transitions mediated by gapless nodes, which we classify as type-(i) and type-(ii) mixed Weyl points, depending on the lattice symmetries.

Following the reviewer's suggestions, we clarified the role of symmetry in the revised manuscript. For the example of the planar model, we emphasize that the symmorphic T*M symmetry is *per se* not protecting the emergent nodal points for an in-plane magnetization direction but it enforces the vanishing of the Chern number. If the system is originally in a non-trivial phase for magnetizations with out-of-plane component, this symmetry thus necessitates a topological phase transition for strictly in-plane magnetization directions, leading to the appearance of type-(i) nodal points.

In addition, we used the example of the buckled model that breaks T*M symmetry to resolve more clearly the roles of spin-orbit coupling and exchange interaction in the revised manuscript. Using the language of the von Neumann-Wigner theorem of level repulsion suggested by the reviewer, we note that the effective Hamiltonian around the linear band crossing is governed by three effective parameters that depend on the crystal momentum, the magnetization direction, and the magnitudes of spin-orbit and exchange interactions. Therefore, tuning these parameters can result in a degenerate point in the spectrum, which implies that the position of the nodal point in (\mathbf{k}, θ) -space is intimately related to spin-orbit coupling and exchange magnitude. In the light of this complex interplay of mixed topology and magnetic interactions, we emphasize in the revised manuscript that the magnetization direction for which the type-(ii) mixed Weyl point emerges is set by the interaction parameters of the model, rather than symmetry. Moreover, we draw a sharp distinction between the type-(i) and type-(ii) nodal points in terms of their minimal number and the associated change of the Chern number.

Finally, we would like to emphasize that the mixed Weyl points in two-dimensional ferromagnets are directly connected to the emergence of the quantum anomalous Hall phase, which has been realized in various theoretical proposals and experimental systems with a wide range of chemical constituents, crystal structures, and magnetic properties in terms of spin and orbital degrees of freedom, see [Ren et al, Rep. Prog. Phys. 79, 066501 (2016)]. In mediating topological phase transitions, mixed Weyl points can consequently be anticipated to emerge as pervasive topological objects for diverse crystal symmetries and orbital chemistry in two-dimensional spin-orbit ferromagnets. Although the current work focuses on systems with primarily p-character of the electrons, we emphasize at the end of the Results section in our revised manuscript that systems with s-orbitals on a bipartite lattice offer similar prospects, owing to the intrinsic valley degree of freedom, which we exploited in our previous work [37].

Reviewer #1: Two materials VOI_2 and Na_2CrBi are interesting, but the feasibility of these materials is essential. For VOI_2 , it seems that the ref 43 only includes VOCl_3 , not VOI_3 . For Na_2CrBi , I wonder if any experimental efforts in growing similar materials.

Reply: We agree with the reviewer that the feasibility of growing the proposed two-dimensional magnets is an important point as it offers exciting prospects for studying the predicted manifestations of complex mixed topology in experiment. As such, realizing these systems shares challenges with the synthesis of quantum anomalous Hall insulators, a class of materials that bears a lot of potential in the context of magnetically controlled topological phase transitions in 2D ferromagnets. In the revised manuscript, we underline the feasibility of growing these layered materials by explicit calculations of the relatively low cleavage energies and the phonon spectra, which show that the proposed structures are in fact energetically and dynamically stable. In addition, we have replaced the reference in question by the new reference [55] that studies VOI_2 as well. Regarding the second material candidate, we are not aware of any experimental efforts to grow specifically Na_2CrBi , but using magnetic doping of Dirac semimetals offers generally a compelling path towards topological magnetic phases. Well beyond the proposal of realistic candidate systems based on material-specific first-principles theory, we predict general physical manifestations of complex mixed topology, which will certainly stimulate experimental studies on the magnetization dependence of 2D topological magnets and their emergent nodal points. In this context, we expect that 2D van der Waals crystals such as the recently discovered room-temperature ferromagnet Fe_3GeTe_2 (see Refs. [45,46] in the revised manuscript) constitute an ideal starting point to observe experimentally the phenomena that we uncover in our work.

Reviewer #1: The experimental proposal about the possibility of giant current-induced orbital Edelstein effect is also interesting. However, the current work only shows the distribution of orbital magnetization in the momentum space. I think a direct calculation of the giant current-induced orbital Edelstein effect based on Ref. 48 and 49 is required to illustrate the experimental feasibility.

Reply: We thank the reviewer for assessing our proposal of large current-induced orbital magnetism due to the emergent nodal points as “*interesting*”. Based on the reviewer’s valuable suggestion, we included in the revised manuscript our explicit results for the orbital Edelstein effect $m_i = \alpha_{ij}E_j$ in the tight-binding model of p-electrons, clearly underlining our initial point. While the equilibrium orbital magnetization is hardly affected by the magnetically induced topological phase transition, the current-induced orbital response is sharply peaked and changes drastically around the transition point (see Fig. 2d-f). As a characteristic imprint of the mixed nodal points on the orbital properties, this result demonstrates the general feasibility of our proposal for tracking the orbital fingerprints of these topological objects in XMCD measurements.

Reviewer #1: A small problem:

In page 7, there is a statement “This observation is well known in molecular physics as well as in band theory of ferromagnets.” I think a citation is required for this statement.

Reply: We thank the reviewer for this comment and included an appropriate citation in the revised manuscript (Ref. [57]).

Reviewer #2: Authors study metallic states occurring at transitions between different topological phases induced by varying the strength and orientation of exchange fields induced by magnetization in two dimensional magnets. When viewed in the combined momentum and exchange field orientation space, such states appear as nodes and/or nodal lines, which carry non-trivial Berry phases. In close analogy to appearance of such metallic states in pure momentum space (so-called Weyl semimetals) the states are termed here as “mixed Weyl semimetals”.

In an earlier work, authors have already pointed out the existence of the mixed Weyl semimetal phase and its role in producing enhanced magneto-electric effects. The main contributions of the present work are: (i) revealing example conditions under which mixed Weyl states arise starting from a topological crystalline/topological insulator phase (ii) finding natural material candidates harboring mixed Weyl semimetals, and (iii) possible manifestations of mixed Weyl semimetals, which include reordering of local orbital magnetization, presence of mixed fermi arcs and additional topological mode-induced torques in chiral magnetic configurations. These findings are supported by reasonable explanations which makes physical sense.

In recent years, there has been quite a bit of interest in the condensed matter community to find nontrivial manifestations of topology for fundamental and technological applications and I thus believe the present study is timely and of interest for Nature communication.

I have few questions and comments before I can recommend publication:

Reply: We appreciate the reviewer’s assessment that “*the present study is timely and of interest for Nature communication*”, and we hope that our response addresses all raised questions adequately such that the reviewer can recommend publication.

Reviewer #2: It is well known that metallic states arise at the boundary of topological phases induced by exchange fields and have been measured experimentally (see for example Science 358, 1311 (2017), and references therein). A major point of the present work is that these states when viewed in the mixed phase space carry a nontrivial topological charge. Although authors present possible indirect measurements of semi metallic nature (like ARPES as a function of magnetization and XMCD, see 2 below for XMCD related question), it will be nice if the authors could present an experimental scheme to directly measure this topological charge?

Reply: We thank the reviewer for this very stimulating question that has in fact far-reaching consequences for the interpretation of Fermi arcs and topological magneto-electric effects in mixed Weyl semimetals. As the reviewer points out correctly, we propose in our manuscript several indirect means such as ARPES and XMCD that could be used to identify the unique signatures of the proposed mixed topological semimetals. Motivated by the reviewer’s comment, we suggest to extract the topological charge of a two-dimensional mixed Weyl point from the measurements of physical currents in experiments that tune the magnetization direction over a small range of angles in the vicinity of the nodal point, which can be achieved, e.g., in FMR-type experiments. The quantized anomalous Hall transport and the pumped currents due to magnetization dynamics rely sensitively on the magnetization direction, which links the unique changes of these Berry-curvature driven phenomena to the topological charge. We added a corresponding remark in the revised manuscript, leaving the precise mathematical formulation for future work.

Reviewer #2: Authors point out that the mixed Weyl nodes arise due to two mechanisms (i) presence of combined time reversal and mirror symmetries, and (ii) interplay of spin-orbit and exchange interactions. Later, they point out in a model system how spin-orbit interaction leads to mixed Weyl points, which are accompanied by a rearrangement of orbital character. Please clarify if the rearrangement of local orbital character is specific to spin-orbit coupling induced mixed Weyl nodes or is universal, i.e. it occurs for both mechanisms (i) and (ii)? This is specially of relevance for the proposed experimental observation of current-induced change in local orbital magnetization, as a signature of mixed Weyl node.

Reply: We thank the reviewer for this question regarding the arrangement of orbital character for the two types of mixed Weyl nodes, which allows us to clarify this important point. First of all, we would like to stress that spin-orbit coupling is a necessary ingredient for the emergence of both type-(i) and type-(ii) nodal points. As we discuss in our manuscript, we distinguish these two types based not only on spin-orbit coupling but we take also into account exchange interactions and symmetries. To address the reviewer’s question on the orbital inversion, we refer to the tight-binding model of p-electrons in the revised manuscript, which can host type-(i) and type-(ii) mixed Weyl points depending on the symmetries of the crystal lattice. Specifically, we refer to Fig. 2d and Supplementary Figure 1b, which demonstrate that the inversion of the orbital character of the energy bands (here in terms of p_x-ip_y and p_x+ip_y states) is a universal signature of both types of mixed Weyl points in orbitally complex systems. This is also visible from Figs. 6d and 6e that illustrate the local variations of the orbital magnetization in the vicinity of type-(ii) and type-(i) nodal points, respectively. Therefore, a large current-induced orbital magnetization should be observable by XMCD for both types of mixed Weyl points. In terms of the orbital Edelstein effect $m_i = \alpha_{ij} E_j$, which we evaluate explicitly for the model in the revised manuscript, however, this implies that the material is gyrotropic, leading to symmetry constraints for a finite linear effect in response to an applied electric field. For instance, this rules out the linear orbital responses α_{zx} and α_{zy} in the planar p-model.

Reviewer #2: The authors show that in “textured magnetic Weyl semimetals” metallic puddles are formed. Additionally they speculate that such electronic puddles are expected to give rise to topological contributions to torques and Hall effects. Similar metallic puddles are known to form in a simple example of a magnetic domain wall exchange coupled to topological insulators (see for example PRL 108, 187201 (2012)) and quantum anomalous Hall insulators [see for example PRB 92, 085416 (2015); PRB 94, 020411(R) (2016)], where topological metallic puddles-induced domain wall motion has also been studied. Naively, it looks like these studied systems are an instance of “textured magnetic Weyl semimetals” proposed by the authors. Can authors comment on this or compare their case with the above mentioned studies?

Reply: We thank the reviewer for drawing our attention to these interesting works, which we missed in the original manuscript. The reviewer is right that the interfaces between topological insulators and dynamic magnetization structures studied in these references could in principle feature a non-trivial mixed topology, analogously to the chiral magnetic textures that we suggest. Therefore, we included these references in the revised manuscript as potential examples of textured topologies. However, we remark that our work has a fundamentally different viewpoint as compared to the suggested references since we study directly the non-trivial topology of such systems in the complex parameter space of crystal momentum and magnetization direction, and provide an interpretation of the metallic puddles in terms of Fermi arcs.

Reviewer #2: On a related note to point 3, according to earlier work by the authors [Nature Comm. 8, 1479 (2017)], even outside the Weyl semimetal phase large non-dissipative torques can appear on application of electric fields. It is not obvious how topological puddles-induced torques can be separated from such contributions. Can authors clarify this point?

Reply: We thank the reviewer for this valuable question, which allows us to elucidate the role of mixed Fermi arcs for magnetization control. Indeed, our previous work (see Ref. [37] in the revised manuscript) demonstrates that mixed Weyl points in the electronic structure result in large anti-damping spin-orbit torques, which enables low-dissipation magnetization control of magnetism by electric fields even in the trivial insulating phase of two-dimensional topological materials. In our current manuscript, we propose that mixed Fermi arcs at the “surface” of textured ribbons of mixed Weyl semimetals (see Fig. 6f in the revised manuscript) can also be exploited for the electric field control of magnetism. In particular, this mechanism could activate field-like spin-orbit torques, the symmetry of which is different from the anti-damping torques that we studied in Ref. [37]. As a consequence, these field-like torques will manifest in very distinct magnetization dynamics that can be used as characteristic probe of the absence or presence of the mixed Fermi arcs in textured ribbons of mixed Weyl semimetals.

Reviewers' comments:

Reviewer #1 (Remarks to the Author):

The updated manuscript is indeed improved and the model discussed at the beginning makes the physics more clear. However, I still have some concerns and confusion about some statements in the current manuscript. Therefore, I still do not think the current manuscript has achieved the high standard of Nature Communications.

1. The authors classify the so-called mixed Weyl points into two categories, type (i) and (ii) as discussed in the manuscript. However, I'm not if these two types are really different. Actually I feel the type-(i) is just a special case of type-(ii). In other words, my question is for type-(i) mixed Weyl point, does it not require the interplay of exchange interaction and spin-orbit coupling? From the model studied in this work, I do not think so.

If I understand correctly (I may have some misunderstanding), the combination of mirror and time reversal requires all the Weyl points appear at $\theta=90$ while the breaking of mirror symmetry will shift the Weyl points to other θ values (I think it is still symmetric with respect to $\theta=90$, right?). If this is correct, that means there is no essential difference between type (i) and type (ii). For type (i), the symmetry gives constraint on the position of Weyl points in the mixed momentum-magnetization direction (θ) space. Actually the symmetry constraint on the position of Weyl points has been studied previously in 3D Weyl systems, e.g. Phys. Rev. Lett. 121, 106402 2018. I think there is deep relation between the current work and the conclusion in this reference.

2. I cannot agree with the statement "As a consequence, we speculate that the nodal points in mixed Weyl semimetals are intrinsically more stable against perturbations of the Hamiltonian ". I think the stability of the Weyl point in the (k, θ) space (2+1) is completely equivalent to the Weyl points in the momentum space (3D).

Reviewer #2 (Remarks to the Author):

The authors have successfully addressed all the concerns in my view. I thus recommended publication.

We would like to thank the two reviewers for taking the time to read our manuscript as well as for providing helpful comments. We thank Reviewer #2 for recommending our work for publication in Nature Communications. In the following, we present a point-by-point response to all issues raised by Reviewer #1.

Reviewer #1: The updated manuscript is indeed improved and the model discussed at the beginning makes the physics more clear. However, I still have some concerns and confusion about some statements in the current manuscript. Therefore, I still do not think the current manuscript has achieved the high standard of Nature Communications.

Reply: We appreciate the reviewer's assessment that we succeeded in improving the clarity of our manuscript based on the valuable suggestions which we received. We hope that we can resolve the remaining concerns and confusion that the reviewer has about some statements such that our manuscript can be recommended for publication in Nature Communications.

Reviewer #1: The authors classify the so-called mixed Weyl points into two categories, type (i) and (ii) as discussed in the manuscript. However, I'm not if these two types are really different. Actually I feel the type-(i) is just a special case of type-(ii). In other words, my question is for type-(i) mixed Weyl point, does it not require the interplay of exchange interaction and spin-orbit coupling? From the model studied in this work, I do not think so.

If I understand correctly (I may have some misunderstanding), the combination of mirror and time reversal requires all the Weyl points appear at $\theta=90$ while the breaking of mirror symmetry will shift the Weyl points to other θ values (I think it is still symmetric with respect to $\theta=90$, right?). If this is correct, that means there is no essential difference between type (i) and type (ii). For type (i), the symmetry gives constraint on the position of Weyl points in the mixed momentum-magnetization direction (θ) space. Actually the symmetry constraint on the position of Weyl points has been studied previously in 3D Weyl systems, e.g. Phys. Rev. Lett. 121, 106402 2018. I think there is deep relation between the current work and the conclusion in this reference.

Reply: We thank the reviewer for these insightful questions, which we use to emphasize the different nature of the two types of mixed Weyl points. First, we note that we study in our work the interplay between magnetism and topology by explicitly including the magnetization direction into the topological analysis of magnetic systems. As a consequence, both exchange interaction and spin-orbit coupling are key ingredients that underlie the emergence of magnetically controlled topological phase transitions mediated by either type-(i) or type-(ii) mixed Weyl points.

The reviewer understands correctly that symmetries can fix the position (and in fact also the minimal number) of the mixed Weyl points, as we emphasize in our manuscript. We refer to this as type-(i) mixed Weyl points, which are enforced to appear at highly symmetric magnetization directions and which are invariant to changes of the finite magnitude of exchange and spin-orbit coupling. As we outline in our manuscript, these type-(i) points appear at least in pairs with the very same chirality. If the protective

symmetries are broken, the group of type-(i) mixed Weyl points splits and individual nodal points move to different positions in the complex phase space in such a way that their distribution is still symmetric around $\theta=90^\circ$. Each of these individual mixed Weyl points appears under a generic magnetization direction that is not related to symmetry but set by the combination of spin-orbit and exchange interactions. In contrast to the type-(i) nodal points that are enforced by symmetry, we classify the described individual mixed Weyl points as type-(ii) nodes, the position of which in the combined phase space is controlled exclusively by the interaction parameters.

The classification scheme of mixed Weyl points that we establish in our manuscript allows us to distinguish between different types of nodal points, which manifest in distinct physical consequences, including the minimal number of nodal points as well as their susceptibility to changes of the electronic structure. Therefore, we are convinced that introducing the concept of type-(i) and type-(ii) mixed Weyl points is a very useful step to classify general properties of topological magnets based on emergent nodal points. Taking the reviewer's concern into account, we have introduced a statement in our revised manuscript that explicitly contrasts the definitions of the two types of nodal points. Here, we make our argument clear by drawing the following analogy: while topological insulators and topological crystalline insulators share common features, the origin of their non-trivial properties is fundamentally different, which is also reflected in the different names associated with these two material classes. Following the same line of thought, we are convinced that *type-(i) and type-(ii) mixed Weyl points present meaningful concepts* in the context of our work on topological magnets. Of course, we agree with the reviewer that by changing the symmetry of the system, it is possible to transform between the two types of nodal points, just like applied strain could turn a topological crystalline insulator into a topological insulator. However, this does not render the underlying naming schemes redundant.

Finally, we would like to thank the reviewer for drawing our attention to the suggested reference that was published during the review process. The work discusses symmetry conditions under which Weyl points of opposite chirality may not annihilate in the momentum space of conventional three-dimensional Weyl semimetals. We agree with the reviewer that there might be a connection to our work, specifically, to the proposed type-(i) mixed Weyl points that mediate topological phase transitions in two-dimensional magnetic systems. Referring to the suggested work, we point out in our revised manuscript that such a formal connection might exist. However, we have to remark that *mixed Weyl points have fundamentally different impact* on experimental measurements. Emerging in two-dimensional ferromagnets for a fixed orientation, these isolated nodal points imprint prominently on, e.g., transport properties since their partners of opposite topological charge are "hidden", thus not being directly accessible. The latter partners are met for distinctly different magnetization directions, which realize *fundamentally different physical systems* (see also our response to point 2). This will result in unique experimental signatures of the bare (i.e. "unpaired") mixed Weyl points in experiments. In sharp contrast, in conventional Weyl semimetals, not individual nodal points but all of them (i.e. including also their partners of opposite charge in addition) are available at the same time in the very same physical system, contributing collectively to transport phenomena. Thus, *mixed Weyl points are distinctly different from conventional Weyl nodes*, even though they might share some formal analogies.

Reviewer #1: I cannot agree with the statement "As a consequence, we speculate that the nodal points in mixed Weyl semimetals are intrinsically more stable against

perturbations of the Hamiltonian ". I think the stability of the Weyl point in the (k,theta) space (2+1) is completely equivalent to the Weyl points in the momentum space (3D).

Reply: We thank the reviewer for this remark which allows us to explain why we arrived at the careful conclusion that the predicted mixed Weyl points could be intrinsically more stable than their cousins in conventional three-dimensional Weyl semimetals. In mixed Weyl semimetals, partners of nodal points with opposite topological charge emerge at different magnetization directions of the ferromagnet, that is, these nodal points are *realized in distinctly different physical systems*. Starting from a system with fixed collinear magnetic configuration, we would have to drastically perturb the Hamiltonian in order to push the nodal points of opposite charge towards each other. Only in this special case, which certainly does not correspond to a ferromagnetic system, the mixed Weyl points of opposite topological charge could appear in a single physical system. In contrast, conventional Weyl semimetals always host in their momentum space pairs of Weyl points with opposite topological charge. Since these pairs are *realized in the very same physical system*, they can annihilate more easily as a consequence of perturbations that bring together the nodal points of opposite charge. Still, we want to acknowledge the reviewer's concern by rephrasing our statement on the stability of nodal points in conventional Weyl semimetals, referring explicitly to the symmetry-related conversion rules of Phys. Rev. Lett. **121**, 106402 (2018).

Reviewer #2: The authors have successfully addressed all the concerns in my view. I thus recommended publication.

Reply: We thank the reviewer for critically reading our manuscript, providing valuable comments, and recommending our work for publication in Nature Communications.

REVIEWERS' COMMENTS:

Reviewer #1 (Remarks to the Author):

1. I actually find the definition in the reply letter seems more clear than that in the paper. According to the reply letter, the "type-(i) mixed Weyl points are enforced to appear at highly symmetric magnetization directions" due to certain symmetry and "are invariant to changes of the finite magnitude of exchange and spin-orbit coupling". In contrast, when the symmetry is broken, the Weyl points appear in a generic position of the mixed space, which requires the fine-tuning of spin-orbit coupling and magnetization amplitude, in addition to magnetization direction. I agree that the type-(i) mixed Weyl points are more convenient for its realization since one only needs to control magnetization direction. But I do not think they are "two fundamentally different mechanisms". As I pointed in the previous report, type-(i) mixed Weyl points are just a special case. I feel that this statement is exaggerated and would suggest changing it to "two different types".

2. In my previous reply, I thought when the author talks about the stability of mixed Weyl points in 3D mixed space (k_x, k_y, θ) , which is equivalent to the conventional Weyl points in the 3D momentum space. After reading the authors' reply, it seems that the author means the stability of nodal points in 2D system (k_x, k_y) with θ chosen to be a certain fixed value (I would not call 2D nodal point as Weyl point since in 2D, we normally only call it Dirac point as in graphene). If this is what the authors mean, I completely disagree with it. For 2D nodal points, there is no need to annihilate two since one can just perturb the magnetization direction (θ) to remove this 2D nodal point. More mathematically, for a single 3D Weyl point, there is no allowed mass term and thus one always needs to annihilate two. However, for the low energy effective Hamiltonian of 2D nodal point, there is one mass term left and thus there is no need to annihilate two and one can just introduce the mass term as the perturbation to gap out the 2D nodal point. Again I feel this statement is exaggerated and would suggest deleting it.

Reviewer #1: I actually find the definition in the reply letter seems more clear than that in the paper. According to the reply letter, the “type-(i) mixed Weyl points are enforced to appear at highly symmetric magnetization directions” due to certain symmetry and “are invariant to changes of the finite magnitude of exchange and spin-orbit coupling”. In contrast, when the symmetry is broken, the Weyl points appear in a generic position of the mixed space, which requires the fine-tuning of spin-orbit coupling and magnetization amplitude, in addition to magnetization direction. I agree that the type-(i) mixed Weyl points are more convenient for its realization since one only needs to control magnetization direction. But I do not think they are “two fundamentally different mechanisms”. As I pointed in the previous report, type-(i) mixed Weyl points are just a special case. I feel that this statement is exaggerated and would suggest changing it to “two different types”.

Reply: On page 3, we followed the Reviewer’s suggestion and changed “*two fundamentally different mechanisms*” into “*two different types*”.

Reviewer #1: In my previous reply, I thought when the author talks about the stability of mixed Weyl points in 3D mixed space (k_x, k_y, θ), which is equivalent to the conventional Weyl points in the 3D momentum space. After reading the authors’ reply, it seems that the author means the stability of nodal points in 2D system (k_x, k_y) with θ chosen to be a certain fixed value (I would not call 2D nodal point as Weyl point since in 2D, we normally only call it Dirac point as in graphene). If this is what the authors mean, I completely disagree with it. For 2D nodal points, there is no need to annihilate two since one can just perturb the magnetization direction (θ) to remove this 2D nodal point. More mathematically, for a single 3D Weyl point, there is no allowed mass term and thus one always needs to annihilate two. However, for the low energy effective Hamiltonian of 2D nodal point, there is one mass term left and thus there is no need to annihilate two and one can just introduce the mass term as the perturbation to gap out the 2D nodal point. Again I feel this statement is exaggerated and would suggest deleting it.

Reply: It seems that the Reviewer might have misunderstood our previous response, where we discuss the nature of mixed Weyl points in the innately 3D combined phase space, and contrast their stability with the one of conventional Weyl nodes. Still, considering the Reviewer’s concern, we toned down the language of the corresponding statement in the revised manuscript. Specifically, we mention that “it might be more difficult to destroy nodal points in mixed Weyl semimetals” by following the logic outlined in our previous response.